# The Nursing Practice Environment and Patients’ Satisfaction with Nursing Care in a Hospital Context

**DOI:** 10.3390/healthcare11131850

**Published:** 2023-06-26

**Authors:** Paula Agostinho, Teresa Potra, Pedro Lucas, Filomena Gaspar

**Affiliations:** 1Unidade Local de Saúde de Castelo Branco, 6000-085 Castelo Branco, Portugal; 2Nursing Research, Innovation and Development Centre of Lisbon (CIDNUR), Escola Superior de Enfermagem de Lisboa, 1600-190 Lisbon, Portugal

**Keywords:** clinical nursing, nursing, hospital, nursing assessment, patient participation

## Abstract

Nursing, being a profession in health, aims to improve the quality of the response to patients’ demands, which have repercussions on the attitudes, behaviors and performance of nurses. Background. The aim of the study was to evaluate the relationships among the nursing practice environment, nurse–patient interactions and patients’ satisfaction with nursing care in a hospital context. Methods. The study applied a descriptive analysis. Based on the initial exploration of the data, we decided to perform a simple linear regression of the dimensions of the scales. Results. The latent variables and interactions between the different dimensions of the three constructs (the nursing practice environment (PES-NWI), nurse–patient interactions (NPIS-22-PT) and patients’ satisfaction in the hospital context (SAPSNC-18)) were submitted to confirmatory analysis. The model was statistically significant, with a good fit with the data (χ2/gl = 128.6/41 (0.000); GFI = 0.900; AGFI = 0.831; TLI = 0.910; CFI = 0.907; RMSEA = 0.102). Conclusions. The study showed favorable rates of overall satisfaction on the part of patients, such as the nurses’ skills in dealing with their illness/health situation, ability to solve problems in a timely manner, responsiveness to patients’ needs and technical competence.

## 1. Introduction

Hospital organizations in the Portuguese context today involve a complex process. The new health policies constitute new challenges, either through the introduction of changes in the structure of the Portuguese National Health System or through the management models, with the aim of increasing the quality of the services provided and reducing the costs. However, nurses and patients are not always involved in the process of change, creating unfavorable environments in relation to change and the future, which may lead to the failure of the process. The reasons for and the questions stimulating this study resulted from observations of the importance of the hospitalized patients’ participation in assessments of their satisfaction with nursing care.

A reference framework was established, where the concepts and assumptions that correspond to the conception of care in nursing were developed, revealing the state of the art of the themes. As major thematic areas, we identified the characteristics of the environment of the nursing practice and the activities of nurses, which included the different interactions of care and were characterized by a set of deeply interdependent technical and human dimensions constituting an inseparable whole. The Quality-Care model© [1] guided the investigation path. This integrates the concept of satisfaction as a result reflecting nursing care that is appropriate to the hospital context and the therapeutic relationship established between the patient and the nurses. Improving the hospital work environment can be a relatively low-cost strategy to improve care and, consequently, the outcomes for patients and nursing professionals [2,3]. In the Portuguese context of hospital nursing, it is a priority to implement measures that can support decision-making by nurse managers, and improve the working conditions in hospital organizations, the quality of nursing care provided and the results for patients [4]. There is strong scientific evidence showing that improvements associated with the practicing environment can be a strategy to improve the quality of hospital care, thus consequently increasing patients’ satisfaction [5,6]. Quality has become more and more important, constituting a constant journey of learning and improvement through the involvement of nurses [7].

Care is a central element of nursing care and indicates the importance of favorable environments [8]. Nursing care is a multidimensional concept, involving a set of dimensions and conditions that describe the attitudes and behaviors related to showing an interest in and respect for the psychological, social and spiritual values of patients [9,10]. This develops as a specific competence in stages through nursing education and clinical practice, in parallel with other nursing competences, such as “acting in a professional manner”, “clinical reasoning in nursing” and “clinical nursing leadership” [11]. Nurse–patient interactions based on care behaviors have been tested by several theorists [9,10,11]. These authors claimed that caring is the essence of nursing and is the key element of nurse–patient interactions for high-quality healthcare. The implementation of nurse–patient interaction models, based on care behaviors in health systems, can improve the work environment, provide a higher level of satisfaction in nurses and patients [12,13], and guarantee a higher level of patients’ safety and quality healthcare [14,15,16].

Regarding patients’ satisfaction with nursing care, in the hospital context, researchers have sought to find the factors that increase it and have focused on the attributes of care that most influence it. Hospitals respond to the health needs of patients with the best possible quality [17,18]. The patients’ satisfaction is an important and legitimate indicator of the quality of healthcare. It is therefore extremely important to define, measure and evaluate the quality of healthcare provided to promote the patients’ satisfaction [19,20].

There is a consensus on the influence of nurses, as health professionals, on the patients’ satisfaction, particularly in the hospital context [21,22]. Compared with other health professionals, nurses spend more time with hospitalized patients and interact with them more frequently, thus having a significant impact on the experience of hospitalization [23,24]. In this way, nurses have the opportunity to approach patients and learn about their expectations [25]. There is a shortage of studies in the Portuguese context, as well as studies that integrate all perspectives and articulate contributions of the patients and nurses, which can broaden our view of the subject.

The study’s main objective aims to assess the existence of relationship between the nursing practice environment, nurse–patient interactions and patient satisfaction with nursing care in a hospital context. Thus, the following research question was defined: What is the relationship between the nursing practice environment, the nurse–patient interactions evaluated by the nurses, and the patient’s satisfaction with nursing care in the hospital context? Specific questions emerge: What is the assessment of the nursing practice environment in a hospital context? What is the assessment of the frequency and importance of nurse–patient interactions, from the perspective of nurses? What is the assessment of patient satisfaction with nursing care in the hospital context? What is the assessment of the relationship between the variables of the nursing practice environment and patient satisfaction in the hospital context? What is the assessment of the relationship between nurse–patient interactions and patient satisfaction in the hospital context?

## 2. Materials and Methods

### 2.1. Research and Design

In this investigation path, we tried to integrate the perspectives of all those involved: nurses and patients. The study had a quantitative, descriptive, correlational, and inferential design. The present study took place in the context of the orthopedic inpatient units and the medicine, surgery, and gynecology/obstetrics departments of four hospitals located in the interior of Portugal. The hospital institutions have between 125 and 170 beds. These hospitals have around 100,000 to 150,000 inhabitants in their catchment area.

### 2.2. Participants

#### 2.2.1. Nurses

Inclusion criteria were applied to select all nurses of both genders according to the service where they carried out their professional activity who had been practicing for over a year and who agreed to participate in the study. 

#### 2.2.2. Patients

The following inclusion criteria were established: informed and voluntary completion of the assessment scale regarding satisfaction with nursing care, being aged over 18 years, being in a physical and psychological condition to participate, hospitalization equal to or greater than 72 h and command of the Portuguese language.

The non-probabilistic sample consisted of 169 nurses and 169 patients. As we did not intend to analyze each service in particular, the data were organized by the hospitals included in the study with the designations Hospital 1, Hospital 2, Hospital 3 and Hospital 4, as shown in Table 1.

### 2.3. Data Collection Instruments

The questionnaire was composed of several measurement instruments. They were the Practice Environment Scale of the Nursing Work Index (PES-NWI) [26], the Nurse–Patient Interaction Scale-22-PT (NPIS-22-PT) [27] and the Patient Satisfaction Assessment Scale with Nursing Care (SAPSNC-18) [28].

The PES-NWI was identified as a sensitive scale to detect differences between different environments of hospital practice, allowing managers to compare the scores of their hospitals with reference values and their impact on the indicators related to the staff and patients [26]. Validation of the Portuguese version of the scale was undertaken by Amaral et al. [26] in the hospital context.

The PES-NWI has 31 items grouped into the following dimensions: participation in hospital policies (9 items); nursing fundamentals for the quality of care (10 items); nurses’ management, leadership, and support skills (5 items); adequacy of human and material resources (4 items); and collegial relationship between doctors and nurses (3 items). From the 31 Likert-type questions (with the lowest value being assigned to “totally disagree” and the highest to “totally agree”, with scores from 1 to 4), a multidimensional scale with five subscales emerged.

In the context of Portuguese nursing, the NPIS-22-PT was translated and validated by Agostinho et al. [27]. It is a relevant instrument that can support the decisions of nurse managers and can improve the working conditions in organizations. The NPIS-22-PT assesses and understands nurses’ perceptions of their interventions in care practice [10] to obtain better health outcomes. With the growing knowledge that nursing performance has an impact on patients’ satisfaction, it has become necessary to analyze and synthesize our knowledge about the relationship between nurse–patient interactions and patient outcomes [27]. The NPIS-22-PT has 22 items grouped by the following dimensions: relational care (9 items), clinical care (8 items) and comfort care (5 items). Items are evaluated for their importance to nurses in providing care using a 5-point Likert scale from “not at all important” to “extremely important”, and for the frequency of providing care using a 5-point Likert scale from “almost never” to “almost always”.

To assess the patients’ satisfaction with nursing care, we used the SAPSNC 18 developed by Freitas et al. [28], thus integrating the perspective of patients. The SAPSNC-18 is composed of three factors and eighteen statements, answered on a Likert scale from 1 to 5 with the statements never to always. The second part of the scale relates to the level of satisfaction with nursing care using a Likert scale ranging from totally dissatisfied to totally satisfied, which allows for assessing the following dimensions: Quality of Care (QC); Quality of Information (QI) and Quality of Service (QS).

### 2.4. Statistical Analysis

We conducted a descriptive analysis and, based on our exploration of the data, we decided to perform a simple linear regression of the dimensions of the scales. We performed descriptive, inferential, and predictive analyses. Simple regression models were used to examine the associations among the nursing practice environment, nurse–patient interactions and patients’ satisfaction with nursing care. To perform the parametric tests, the assumptions of normality were verified. In all statistical tests, a maximum permissible error of 5% (α = 0.05) was assumed as a decision rule [29]. We evaluated the Cronbach’s alpha of the dimensions as a way to assess the internal consistency of the constructs. In the statistical analysis, IBM Statistical Package for the Social Sciences (SPSS) (version 24) and IBM SPSS AMOS^®^ (version 22) were used.

### 2.5. Ethical Procedures

We obtained authorization from the authors of the scales we used in the present study. The study was authorized by the boards of directors and the ethics committees of the health institutions (opinion No. 201901206, 12 June 2019). The study was conducted in accordance with the ethical standards of the Declaration of Helsinki [30] and in accordance with the legal regulations [31].

## 3. Results

### 3.1. Participants

#### 3.1.1. Patients

Of the 169 patients that made up the sample in this study, about 63.9% (n = 108) were female and about 36.9% (n = 61) were male, with a strong predominance of females, as shown in Table 2.

Regarding age, the youngest individual was 21 years old and the oldest was 91 years old. With regard to the four hospitals, the most represented age groups corresponded to those between 20 and 29 years old (19.0%), between 30 and 39 years old (17.1%) and between 70 and 79 years old (17.0%). On the other hand, the age groups that had smaller percentages were 40–49 years old, 50–59 years old (11.0% each), and 80–91 years old (12.0%), as shown in Table 3.

Regarding education, 2.9% of the patients had no academic qualifications, 7.1% had completed fourth grade and only 9 (5.3%) had higher qualifications. This result indicates that around 33% had qualifications below the level of compulsory schooling (i.e., these patients were elderly and with little schooling). Regarding employment status, 53 (31.4%) patients were retired, 21 (12.4%) were unemployed, 7 (4.1%) were students and 83 (49.1%) were from other professions, and only 2 (1.2%) worked in the field of health. Regarding marital status, most patients (101, 59.8%) were married or in a de facto union, 21 (12.4%) were widowed, 40 (23.7%) were single and 6 (3.6%) were divorced. Regarding the need for nursing care after discharge, 61 (36.1%) of the patients needed nursing care after discharge, which indicated the patients’ disease burden and the need for follow-up in the community, as shown in Table 4.

#### 3.1.2. Nurses

The nurses were taken from a randomized subsample of the study, selected by length of service, yielding a total of 169 participants, the majority of whom (61.5%) were women, as shown in Table 5.

Regarding academic qualifications, the predominant proportion held a degree (33.7%, n = 57). The percentage of nurses with a master’s degree was 15.3% (n = 26). Regarding the professional categories, it was observed that 47 nurses had a specialist title, 21 were nurse managers and most of the respondents were in the category of nurses (101, 59.7%). As for the length of professional practice, it is noteworthy that 66 nurses had been in the profession for over 30 years. Regarding current service, 108 nurses had been at their current institution for over 30 years, as shown in Table 6.

### 3.2. Assessment of the Nursing Practice Environment

Regarding the nursing practice environment in the hospital context, we used the PES-NWI, which consists of 31 items, grouped into five dimensions. In general, the answers were distributed between 1 and 4, covering the entire scope of the scale.

The reliability of the PES-NWI scale was evaluated by calculating the Cronbach’s alpha as the coefficient of internal consistency. This ranged from a minimum value of 0.72 (acceptable) for the dimensions of management capacity, leadership, and support of nurses, to a maximum value of 0.81 (good) for the relationship between physicians and nurses, revealing the good internal consistency of the dimensions. At the global scale, the study had very good internal consistency, with a Cronbach’s alpha of 0.83, as shown in Table 1.

The dimension of the fundamentals of nursing for quality of care was classified as the most favorable, achieving an average of 2.72 (SD = 0.42), followed by the dimensions of the relationship between physicians and nurses, and the capacity for management, leadership, and support of nurses, with averages of 2.55. The dimension of the participation of nurses in hospital governance ($\bar{x}$ = 2.16; SD = 0.45) and the dimension of the adequacy of human and material resources ($\bar{x}$ = 2.04; SD = 0.56) were considered the most unfavorable by nurses, as demonstrated in Table 7.

### 3.3. Evaluation of Nurse–Patient Interactions

Regarding the results of the Nurse–Patient Interaction Scale-22-PT (NPIS-22-PT), which is composed of 22 items grouped into three dimensions [27] and calculates the average of the two components (importance and frequency), through a general analysis, it was verified that the items varied between 2.50 and 5.

The dimension of comfort care had the highest average importance ($\bar{x}$ = 4.85; SD = 1.36) of the dimensions of the NPIS-22-PT. The dimension of relational care ($\bar{x}$ = 3.75; SD = 0.64) obtained the lowest average. The reliability of the NPIS-22-PT scale was evaluated by calculating the Cronbach’s alpha as the coefficient of internal consistency, which presented acceptable values of reliability (α = 0.86) for the total score for each of the two questions asked (importance and frequency). The values of Cronbach’s alpha for the three components varied across the three dimensions, as shown in Table 8: relational care, α = 0.91; clinical care, α = 0.82; comfort care, α = 0.74. All the results were good [29].

### 3.4. Assessment of Patients’ Satisfaction with Nursing Care in the Hospital Context

The assessment of patients’ satisfaction with nursing care in the hospital context (SAPSNC-18) consisted of 18 items [28] grouped into three factors, and the reliability was assessed by calculating the coefficient of internal consistency (Cronbach’s alpha). Factor 1 (quality of care) presented an α of 0.80, Factor 2 (quality of information) showed an α of 0.78, and Factor 3 (quality of care) presented an α of 0.63. At the global scale, it had very good internal consistency, with a Cronbach’s alpha of 0.90.

From the responses to the SAPSNC-18, an overall average of 4.24 (SD = 0.65) was obtained, showing satisfaction on the part of patients in the hospital context, since the average corresponded to the average value of the Likert scale, which ranged from 1 to 5. It can be seen that the trend of the responses was above the midpoint. The factors of the SAPSNC-18 with the highest average were quality of service ($\bar{x}$ = 4.50; SD = 0.49) and quality of care ($\bar{x}$ = 4.47; SD = 0.47), whereas quality of information ($\bar{x}$ = 3.74; SD = 1.01) achieved the lowest average, according to Table 9.

### 3.5. Relationship between the Nursing Practice Environment and Nurse–Patient Interactions with Patients’ Satisfaction

With regard to the regression models using the PES-NWI and NPIS-22-PT scales with the respective dimensions as simultaneous predictors, with the dependent variable being patients’ satisfaction in the hospital context (SAPSNC-18), it is noteworthy that the predictive variables integrated in the model had a positive relationship that was statistically significant (F = 3.305; *p* = 0.002) with the dependent variable, namely, quality of care; an R^2^ value of 0.099 was also noted. 

The dependent variable (quality of care) was explained by the dimensions of nursing fundamentals for quality of care (β = 0.285; *p* = 0.029); management capacity, leadership, and support for nurses (β = 0.190; *p* = 0.050); and collegial relationships between physicians and nurses (β = 0.203; *p* = 0.010), for which the values were positive and statistically significant. It should be noted that the dependent variable (quality of care) was explained by relational care (β = 0.302; *p* = 0.001), which had a positive value that was statistically significant. Clinical and relational care were not statistically significant.

In the linear regression analysis, the model was statistically significant (F = 2.08; *p* = 0.040), with an R^2^ value of 0.094. It should be noted that the dependent variable of quality of information was explained by the participation of nurses in hospital governance (β = 0.062; *p* = 0.010) and the fundamentals of nursing for the quality of care (β = 0.285; *p* = 0.003), for which the values were positive and statistically significant.

In the linear regression analysis, the model was not positive and statistically significant (F = 1.95; *p* = 0.056), with an R^2^ of 0.089. It should be noted that the dependent variable of quality of care was explained by the collegial relationships between doctors and nurses (β = 0.271; *p* = 0.021) and by relational care (β = 0.186; *p* = 0.040), for which the values were positive and statistically significant.

In summary, a graphical representation of the final model with the latent variables is presented in Figure 1.

The model underlying the research question of the study was submitted to confirmatory analysis (path analysis) and a reasonable adjustment was verified, confirming the latent variables and the interactions among the different dimensions (as shown in Figure 1) of the three structures: nursing practice environment (PES-NWI), nurse–patient interactions (NPIS-22-PT) and patients’ satisfaction in the hospital context (SAPSNC-18). The model was statistically significant, with a good fit to the data (χ^2^/gl = 128.6/41 (0.000); GFI = 0.900; AGFI = 0.831; TLI = 0.910; CFI = 0.907; RMSEA = 0.102).

The ideal value of the RMSEA (root mean square error of approximation) is between 0.05 and 0.08; however, values of up to 0.10 are accepted, particularly in exploratory studies [29]. It should be noted that the adjustment becomes even more difficult when different sources of data are involved, as was the case here. In the synthesis model presented here, the PES-NWI and NPIS-22-PT scales were evaluated by nurses and the SAPSNC-18 was evaluated by the patients. The NPIS-22-PT had a much higher predictive value. As a result, clients are, in a global way, satisfied with nurses’ skills to deal with their disease/health situation and the ability to solve problems in a timely manner, as well as responsiveness to their needs and technical competence. The clients considered that nurses sometimes transmit the usefulness of information during hospitalization and after discharge, and in some circumstances provide clients with the necessary skills to handle their disease/health care. Regarding the association between variables, the relevance of relational care as a predictor of satisfaction with nursing care is verified, as regards the practice environment as a predictor of customer satisfaction and the participation of nurses in the management of the hospital; this was also seen for nursing foundations for the quality of care, management capacity, leadership and support of nurses, and collegialrelationships between physicians and nurses.

## 4. Discussion

The study, applied to 169 nurses and 169 patients from the four hospitals mentioned above, has as its general objective to evaluate the relationship of the nursing practice environment, nurse–patient interactions, and patient satisfaction with nursing care in a hospital context. From the evaluation of the nursing practice environment in a hospital context, although the number of participants is lower than in other studies, the values obtained are close to most of the investigations already carried out in Portugal [26,32]. Studies conducted in other countries show frankly more favorable values [5,33,34,35]. We can affirm that the dimension fundamentals of nursing for the quality of care obtained a better classification by nurses, having been positive in all hospitals. These results are, however, lower than those of studies in Portugal [26,36,37] and at the international level, with similar results [25,38,39]. This dimension, when effective, shows a lower mortality rate, higher job satisfaction, improvement in the quality of care and fewer adverse events [40,41]. Regarding the dimension of the relationship between physicians and nurses, a favorable overall score was obtained in the present investigation, which was verified in all the hospitals of the present study. This dimension is in line with the studies of [26,32,42], and [37] showed a negative average for this size. A work environment with a favorable relationship between nurses and physicians is fundamental for the success of the care that occurs in a positive work environment and favorable practice [36]. At the international level, there were superior results [38,39,42]. According to the literature, a good relational level between nurses and physicians leads to greater professional satisfaction and reduces the rates of intention to leave the service, and improves the climate of support among professionals [43,44]. It should also be noted that physician–nurse collaboration and teamwork can improve patient outcomes and reduce healthcare costs [45,46], increase job satisfaction and maintain patient safety [4,47,48]. The dimension adequacy of human and material resources and the participation of nurses in the governance of the hospital obtained a lower average rating. Referring to the perception of nurses regarding existing resources affected by the number of nurses present in the unit, but also by the number of beds and typology of patients, all hospitals in the present study presented unfavorable results. Similar results were found at the national level [26,32,37]. At the international level, similar results [6,38,39] were found to the present study. These results demonstrate that this is one of the most sensitive areas of the nursing practice environment, and that without sufficient resources, nurses have more difficulty in providing care of excellence, which has been recurrent and problematic at the level of the National Health System. This result, in relation to the dimension of the adequacy of human and material resources, may probably be associated with the nurse/patient ratio in the services, in which the high complexity of the typology of patients requires nurses to be more vigilant and permanent. This problem rates to the existence of an inadequate nurse/patient ratio in most hospitals in Portugal [26,32,37]. The high and representative work overload of Portuguese nurses and the calculation of the needs of nursing endowment cannot be limited to the criterion of the number of hours of care per patient and per day, or average times used in certain procedures [49]. It is critical that hospital administrators find solutions to provide safer environments [49]. Regarding the dimension of nurses’ participation in hospital governance, it presents a lower average rating, as does nurses’ perception of the importance and opportunity to participate in political decisions. Similar results are found at the national level [26,32] and the international level [50,51]. This raises an opportunity for professional appreciation and the sharing of concerns [36]. The involvement of nurses in organization and decision-making is considered a good practice and contributes to good results [36]. The constant transformations and advances in the scenarios of health practices, especially in the hospital environment, have had repercussions on how the teams organize themselves for the provision of care. This situation requires from professionals, especially nurses, specialized knowledge in management, as well as experience in the area of administration [2,52], allowing them to act with more autonomy, as well as interfere in decision-making within the organization [2]. Shared management and involvement in the decision-making process that involves nursing practice increases commitment to the profession, as well as the sense of empowerment and professional satisfaction [53,54]. Involvement in the strategic decisions of the organization is related to feelings of satisfaction, emotional exhaustion, and intention to leave the profession, and can help managers find ways to promote the well-being of their professionals, namely, by providing support and effective responses to the concerns and daily problems of nurses [2,50]. The possibility of nurses getting involved and participating in the policies and issues of the hospital may represent an opportunity for professional appreciation and sharing of concerns [36]. The results regarding the dimension of management, leadership and support capacity for nurses obtained a favorable global score; however, there is a differential between the hospitals. In the comparison, results are higher than those presented by [32,37], where they identified a negative overall score. Refs. [26,36] recorded a higher global average. In international studies, we found results like those of the present study [6,38,39,42].

The dimension of the NPIS-22-PT that presented the highest overall mean value in terms of importance was comfort care. This dimension indicates how nurses perceive the effectiveness of the process and evaluate aspects related to interactions in the treatment regarding the transmission of information in an understandable way, listening skills, the ability to solve problems in a timely manner, the ability to respond to patients’ needs, and technical competence. The dimension of relational care had the lowest mean value for importance and frequency. According to the perceptions of the nurses, this revealed that the nurses evaluated the aspects related to the challenges perceived by the patients as considerably important. Favorable nurse–patient interactions reduced the days of hospitalization and improved the quality of patients’ and nurses’ satisfaction [8,35]. Regarding the clinical care dimension, it presented a favorable average within the total scale (importance and frequency). Nurses’ attitudes influence, and are influenced by, the functional capacity for self-care, self-esteem, and satisfaction with life. These allow us to evaluate aspects related to self-care behavior; this can be associated with personal attitudes towards others and the future. This attitude influences the way nurses perceive the health condition, interact with patients, and position themselves before the negotiated treatment [10,13,16].

With regard specifically to the assessment of satisfaction, we opted for the SAPSNC-18 scale validated in 2016 in Portugal, although no subsequent national or international studies are known. The SAPSNC-18 considers the perceptions of patients in relation to their satisfaction with the care received, that is, the care that patients experienced. The existing literature reveals that, in most studies of patient satisfaction, it has favorable overall satisfaction indexes [19,21,28], which corroborates the results of the present study. The dimension that obtained the greatest satisfaction on the part of patients was the quality of care, which is higher than that in the study by [28]. Scientific evidence reveals that the assessment of the physical environment, the relationships between professionals and patients, communication [19,22], privacy and security [18] and the provision of information about the care provided [12] are more relevant factors affecting the perceptions of patients. Regarding the quality of care dimension, it obtained an average higher satisfaction score on the part of patients, according to the study by [28]. Nurses spend more time with hospitalized patients and interact more frequently with them, so they have a significant impact on the quality of patient care and the hospitalization experience [18,21]. The dimension that obtained the lowest satisfaction on the part of patients was the quality of information, which is like what was found in the study by [28]. The existing literature reveals levels of dissatisfaction in the perceptions of patients when there is a greater lack of continuity in relation to healthcare [14,21].

Regarding the relationships of the variables of the nursing practice environment and nurse–patient interactions with patients’ satisfaction in the hospital context, we found that the nurses’ perceptions of the working relationships with physicians, nursing care plans, continuing education and the recognition of work had a positive relationship with how patients perceived the effectiveness of the communication process and the effectiveness of personalized care. These findings are in line with other evidence [55,56], being a predictor of overall satisfaction with the healthcare experience, particularly the way nursing care is provided [57]. It is also agreed that there is a relationship between satisfaction with the health service as regards expectations and perceptions about the care received, that is, the care that the client experienced and what they expected to receive [56]. In this specific context and according to the scientific literature, there is a growing recognition that quality health services must be effective, safe, and people-centered [3,5]. In the study by [20], they reported care as a special focus in therapeutic relationships since they translate into expressions of the interaction of one or more people. Promoting the resolution of problems is encouraged so that the patient finds alternative ways of seeing the situation, meeting their expectations, and seeking recognition and satisfaction. Refs. [1,11] argue that the attitudes and behaviors that nurses put into practice in partnership with patients and families are facilitators of specific interventions in the disciplinary area of nursing, and lead to results “highly” determinative of nursing care. It is important to look at the patients as a relational experience regardless of the situations that occur, whether in health or in disease [10].

According to the perceptions of nurses in this study regarding their participation in the hospital’s internal governance, several indicators, such as continuity of care and the use of diagnoses and nursing care plans, had a positive relationship with the usefulness of the information transmitted to the patient during hospitalization and after discharge. This information aims to provide the patient with the necessary skills to deal with their illness/health situation, according to a study [57]. In the scientific literature, we find other studies that confirm that the practice environment is composed of the accumulated effects of numerous interrelated factors, which affect performance and satisfaction [32]. Others [16] point out that the nurse–patient relationship/interaction has been described as a process of interpersonal interaction via self-care strategies, medication adherence and psychological interventions. Moreover, they refer to the relational dimension as an attitude of improvement in patient care [12].

Patient satisfaction with nursing care evidence that patients at the time of hospital discharge experienced a positive relationship between the confidentiality of information, care, treatments, the administration of medications in a timely manner, safety during hospitalization and satisfaction with nursing care [21]. In fact, satisfaction with nursing care has been systematically discussed in the scientific literature as a predictor of overall satisfaction with the healthcare experience [20,21,22].

It is expected that the development of this research will contribute to the construction of affinity, based on the knowledge of the individual who is cared for, and the deepening of relationships and bonding between those involved. It is noteworthy that this study can help in the provision of care by the nursing team and strengthen the relationship between professionals and clients to achieve better health outcomes. On the other hand, it could contribute as a strategic measure to improve the clinical and organizational quality of health institutions, by increasing accountability for the quality of all levels of the system and increasing the involvement of professionals and leaders.

### Limitations and Suggestions

Despite the methodological options chosen, which considered methodological rigor, the constituents of the sample, the evaluation of psychometric properties and the use of statistical tools such as linear regression models, this study had some limitations. For example, the various instruments had different scales: the PES-NWI had a Likert scale from 1 to 4, and the NPIS-22-PT and SAPSNC-18 had scales from 1 to 5. Secondly, the origin of the data made the robustness of the models more difficult to determine, which in this case did not happen, because the final model obtained by the confirmatory analysis had a marginal RMSEA. The integration of the patients’ voices in research studies can also reinforce knowledge and partnership, and improve the focus on the person regarding what is desired, as a model of care where the patient holds power and autonomy over his/her health process/illness. In fact, these constraints are understood as challenges and suggestions for the development of new studies.

Another limitation of the study was the timing, as it took place at the beginning of the pandemic, which may have led to greater dissatisfaction on the part of the participants, as adverse conditions occurred that had not previously been experienced either by patients or by nurses.

Another limitation of the study to be mentioned is the fact that the participants’ responses were based on their perceptions, which may have been affected by the social desirability of patients, with the responses being biased by the tendency of individuals to seek social approval and avoid criticism, according to what is culturally acceptable.

Although this study has specifically focused on nursing care, it is difficult to separate this factor from the complexity of healthcare that hospitalization entails. In this sense, it is important to analyze satisfaction in a dynamic way, considering the various disciplinary areas involved. A study has been proposed to consider the dimensions of satisfaction related to other health professionals as well as those in relation to nursing care, as each one influences satisfaction. On the other hand, different strategic measures must be developed to improve the clinical and organizational quality of health institutions by increasing accountability for the quality at all levels of the system, and by increasing the involvement of professionals and leaders.

Finally, as a suggestion for the development of future research, studies could assess clients in other age groups to understand the associations of age and developmental stage with satisfaction with hospitalization.

## 5. Conclusions

The nurses were satisfied with their work and with the established relationships, but they considered the available resources to be scarce, and that participation in hospital policies and the support of management and leadership were not favorable for them to continue to develop their functions.

This study has demonstrated that in the assessment of nurse–patient interactions, nurses evaluate attributes related to the interaction during the treatment, interact with the patients, and perceive the importance and frequency of the nurse–patient relationship via which nursing care is developed.

Given the results, models for the planning and management of environments, resources and care can be restructured in a reasoned way, and greater participation of nurses in hospital governance is also required, as well as adequacy of human and material resources, and effective management, leadership, and support of nurses, to achieve better levels of quality and patient satisfaction. As well, different strategic measures must be developed to improve the clinical and organizational quality of health institutions, through increased accountability for the quality of all levels of the system and the increased involvement of professionals and leaders. For professionals involved in the direct provision of care, with conditions inherent to the nursing practice environment (professional practice), the results of this study contribute to the valorization of nursing.

## Figures and Tables

**Figure 1 healthcare-11-01850-f001:**
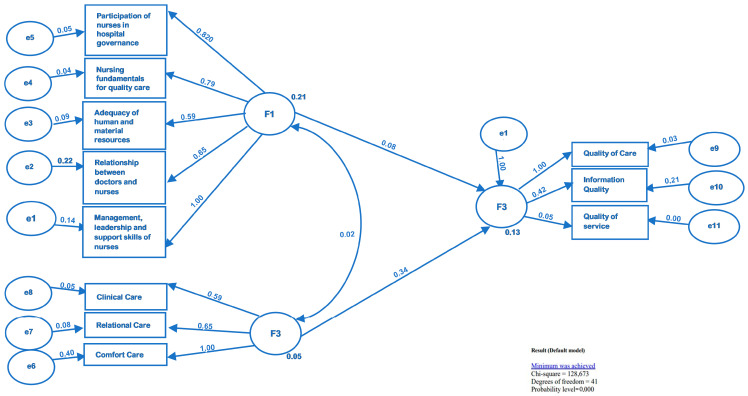
Final model with latent variables.

**Table 1 healthcare-11-01850-t001:** Application of questionnaires to nurses and patients in the hospital context.

	Nurses	Patients
Hospital 1		
Medicine	26	20
Orthopedics	15	11
Surgery	12	8
Gynecology/Obstetrics	10	15
Hospital 2		
Medicine	29	28
Orthopedics	11	11
Surgery	9	10
Hospital 3		
Medicine	15	11
Orthopedics	7	11
Surgery	10	11
Gynecology/Obstetrics	6	21
Hospital 4		
Medicine	19	12
Total	169	169

**Table 2 healthcare-11-01850-t002:** Sample distribution (n = 169) by institution, according to the patients’ gender.

Patients						
	Male		Female		Total	
	n	%	n	%	n	%
Hospital 1	18	10.6	36	21.3	54	31.9
Hospital 2	28	16.6	21	12.4	49	28.9
Hospital 3	10	5.9	44	26.0	54	31.9
Hospital 4	5	2.9	7	4.1	12	7.1
Total	61	36.9	108	63.9	169	100

**Table 3 healthcare-11-01850-t003:** Distribution of each age group of patients in the four hospitals.

	Age Group						
	20–29	30–39	40–49	50–59	60–69	70–79	80–91
Hospital 1	11	10	4	3	5	12	9
Hospital 2	5	5	8	9	11	7	4
Hospital 3	14	14	6	5	6	6	3
Hospital 4	2	0	0	1	1	3	5
Total	32	29	18	18	23	28	21
	19.0%	17.1%	11.0%	11.0%	13.6%	17.0%	12.0%

**Table 4 healthcare-11-01850-t004:** Distribution of the sample by hospital, according to educational qualifications, marital status, work situation and need for nursing care after discharge.

Patients											
			Hospital 1		Hospital 2		Hospital 3		Hospital 4		
			*n*	*%*	*n*	*%*	*n*	*%*	*n*	*%*	Total
Academic qualifications	Without qualifications		5	2.9	4	2.3	0	0	3	1.7	12
	Fourth grade		12	7.1	15	8.8	9	5.3	3	1.7	39
	Ninth grade		7	4.1	10	5.9	7	4.1	2	1.1	26
	Twelfth grade		21	12.4	16	9.4	16	9.4	4	2.3	57
	Graduate		9	5.3	4	2.3	22	13.1	0	0	35
Marital status	Single		12	7.1	9	5.3	17	10.0	2	1.1	40
	Married		31	18.3	33	19.5	32	18.9	5	2.9	101
	Divorced		2	1.1	4	2.3	1	0.5	0	0	7
	Widower		9	5.3	3	1.7	4	2.3	5	2.9	21
Employment status	Retired		22	13.1	13	7.6	9	5.3	9	5.3	53
	Domestic		2	1.1	0	0	1	0.5	0	0	3
	Unemployed		5	2.9	6	3.5	10	5.9	0	0	21
	Student		3	1.7	1	0.5	1	0.5	2	1.1	7
	In the health area		0	0	2	1.1	0	0	0	0	2
	Others		22	13.1	27	15.9	33	19.5	1	0.5	83
Need for nursing care after discharge	Yes	Treatment	11	6.5	12	7.1	6	3.5	0	0	29
		Injectables	6	3.5	1	0.5	6	3.5	3	1.7	16
		Home visits	7	4.1	4	2.3	2	1.1	3	1.7	16
	No		30	17.7	32	18.9	40	23.6	6	3.5	108
Total hospitals			54	20.5	49	18.6	54	20.5	12	4.6	169

**Table 5 healthcare-11-01850-t005:** Sample distribution (n = 169) by institution, according to the nurses’ gender.

Nurses						
	Male		Female		Total	
	n	%	n	%	n	%
Hospital 1	22	39.9	41	65.0	63	37.2
Hospital 2	21	42.8	28	57.1	49	28.9
Hospital 3	13	34.2	25	65.7	38	22.4
Hospital 4	8	42.1	11	57.8	19	11.2
Total	61	36.0	105	62.1	169	100

**Table 6 healthcare-11-01850-t006:** Sociodemographic characteristics of the nurses.

Nurses										
		Hospital 1		Hospital 2		Hospital 3		Hospital 4		
		*n*	*%*	*n*	*%*	*n*	*%*	*n*	*%*	Total
Academic qualifications	Bachelor’s degree	21	12.4	16	9.4	16	9.4	4	2.3	57
	Graduate	12	7.1	15	8.8	9	5.3	3	1.7	39
	Master’s degree	7	4.1	10	5.9	7	4.1	2	1.1	26
	Specialization	5	2.9	4	2.3	0	0	3	1.7	12
	Postgraduate	9	5.3	4	2.3	22	13.0	0	0	35
Professional category	Nurse	31	18.3	33	19.5	32	18.9	5	2.9	101
	Specialist nurse	14	1.1	13	2.3	18	0.5	2	1.1	47
	Nurse manager	9	5.3	3	1.7	4	2.3	5	2.9	21
Professional activity (years)	0–9	22	13.0	13	7.6	9	5.3	9	5.3	53
	10–19	12	7.1	9	5.3	3	1.7	5	2.9	29
	20–29	5	2.9	6	3.5	8	4.7	2	1.1	21
	30–39	39	23.0	13	7.6	11	6.5	3	1.7	66
Professional activity in the current organization (years)	0–9	11	6.5	12	7.1	6	3.5	0	0	29
	10–19	6	3.5	1	0.5	6	3.5	3	1.7	16
	20–29	7	4.1	4	2.3	2	1.1	3	1.7	16
	30–39	30	17.7	32	18.9	40	23.6	6	3.5	108
Total hospitals		63	37.2	49	28.9	38	22.4	19	11.2	169

**Table 7 healthcare-11-01850-t007:** Descriptive analysis of the PES-NWI by dimensions in the hospital context.

Dimensions	$\bar{x}$	SD	α
Participation of nurses in hospital governance	2.16	0.45	0.80
Nursing fundamentals for quality care	2.72	0.42	0.78
Adequacy of human and material resources	2.04	0.56	0.78
Relationship between doctors and nurses	2.55	0.57	0.81
Management, leadership, and support skills of nurses	2.55	0.43	0.72
PES-NWI	2.42	0.37	0.83

$\bar{x}$, mean; SD, standard deviation; α, Cronbach’s alpha.

**Table 8 healthcare-11-01850-t008:** Descriptive analysis of the NPIS-22-PT by dimensions (frequency and importance) in the hospital context.

Dimensions	$\bar{x}$	SD	α Importance	α Frequency	α Global
Clinical care	4.17	0.51	0.77	0.81	0.91
Relational care	3.75	0.64	0.91	0.93	0.82
Comfort care	4.33	0.46	0.76	0.70	0.74

$\bar{x}$, mean; SD, standard deviation; α, Cronbach’s alpha.

**Table 9 healthcare-11-01850-t009:** Descriptive analysis of the factors of the SAPSNC-18.

Dimensions	$\bar{x}$	SD	α
Quality of care	4.47	0.47	0.80
Information quality	3.74	1.01	0.78
Quality of service	4.50	0.49	0.63

$\bar{x}$, mean; SD, standard deviation; α, Cronbach’s alpha.

## Data Availability

The data presented in this study are available on request from the corresponding author.

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
