# Peer review of "The Nursing Practice Environment and Patients’ Satisfaction with Nursing Care in a Hospital Context"

_healthcare, 2023, doi:10.3390/healthcare11131850_

Round 1

Reviewer 1 Report (New Reviewer)

1. In the introduction add main objective or research question.

2. Add literature review section or instead of review, add hypotheses development section and relevant theory and develop relevant hypotheses

3. In the Abstract and resutls section RMSEA is 0.102, it must be less than 0.08.

4. Figure 1 shows that following variables (Management,leadership and
support skills of nurses, comfort care and quality care) have value of 1 which is over estimated if this issue is resolved then might be RMSEA would come in the range.

5. Moreover, information quality and quality of servcie have loadings less than 0.5 hence this needs attention.

6. Justify hypotheses in discussion and relate them to past studies.

7. Add Discriminant validity table

8. It is better to add confirmatory factor analysis (CFA) model and results then add simple linear regression results.

Spelling and gramamr should be checked before publication of this paper.

Author Response

Reviewer 2 Report (New Reviewer)

Thank you for the opportunity to review your manuscript.

Your manuscript reads well overall but can be further improved by addressing the comments highlighted below.

My main concern is that data from two different samples (nurses and patients) are collected and analysed in a single analysis. I’m not familiar with such an analysis, perhaps would be helpful to get a statistician’s input.

Other feedback for consideration:

Introduction

·       The specific questions mentioned in lines 87-93, relate to “how”, they are not suitable to be answered using quantitative study.

Methods

·       It is not clear how sample size adequacy was determined in this study, what sampling approach was used and how sample matching between nurses and patients was done.

·       Please note that exclusion criteria are not the opposite of inclusion criteria (lines 108-110).

Results

·       I do not see the need for presenting the patients' demographic data in four separate tables, which can be combined into one table.

·       Consider highlighting the key results in your survey in the text and referencing the tables without repeating the facts.

·       What is the overall perception of the NPE and Nurse–Patient Interactions in this study?

·       You have mainly given the statistical values ​​in the results without corresponding interpretations, especially in the context of the analysis in Figure 1, the readers have to do a lot of work to understand the meaning of the results.

·       Since four different settings are sampled, are there significant differences/similarities in the results of the variables between settings?

·       Are there significant differences/similarities in the outcomes of the variables between the settings since four different settings are being selected?

·       To check the clarity of sentences on results presented in lines 290-291.

·       The results are presented in nine tables. Please reduce this to an acceptable number (ideally a maximum of 5 to 6). For better clarity, the results of the linear regression should be presented in tabular form.

 Discussion

·       The discussion should first focus on the overall results of the variables before focusing on their subscales. \

·       Although the results of the current study were compared to previous studies, this should lead to a more in-depth discussion of the possible causes of the results.

·       It is not clear from the discussion what implications your results have for practice, particularly in relation to the main outcome/study aims.

·       The discussion should be considerably revised to reduce the description and emphasize the implications for practice.

 Conclusion

·       Consider emphasising the importance of your findings for practical application and future studies. What new knowledge does this study add?  

Others

·       Please check for typos and missing/incomplete sentences.

I hope these comments are helpful in considering your manuscript.

Author Response

Reviewer 3 Report (New Reviewer)

Although the scope of the study is regional and the sample is not representative, I consider it significant, especially at the time of global shortage of nurses.

I recommend to the authors:

·      to reduce the number of tabs with the characteristics of the sample,

·      to check the table 6 - two lines in the part Academic qualifications provide information about bachelor´s degree,

·      to more concrete the recommendations for practice.

Author Response

This manuscript is a resubmission of an earlier submission. The following is a list of the peer review reports and author responses from that submission.

Round 1

Reviewer 1 Report

Thank you for the opportunity to review the manuscript on nursing-patient care satisfaction.  The authors did a great job in the design and writing of the manuscript.  Feedback as follows.

The introductory sentence in the abstract appears as a run on and was difficult for the reader to follow without reading more than once.   On the body of the manuscript, the words "patient" and "client" appear to be used interchangeable.  I recommend selecting one for consistency.  Line 119 "In this sense your opinion..." can be confusing as to whom "you" refers to.  

Although the participant demographics and characteristics are summarized in paragraph form, I recommend creating a table for ease of reading.  

The core concept of the study is nursing care.  I recommend defining nursing care by perhaps referencing a nursing care theory/model and extracting the care dimensions measured, such as communication, empathy, of the basic knowledge, skills, and attitudes that reflect the professional nurse.  If described in the manuscript, it was not evident to the reader.  I recommend expanding on this. 

Thank you.

Thank you.  

Author Response

Consulte o anexo.

Reviewer 2 Report

Dear authors,

the subject for your study is important, however, I have som serious problems understanding your choices, arguments and writing.

1. Abstract: I do not understand the lines 9-12. What are you point here?

2. Background. Line 54-56. I do not understand why you mean that satisfied patients are loyal to the hospital etc. Can you explain this further and think about the ethically consequenses. Is this really the reason for why we want satisfied patients?

3. Material and methods. Line 83. The 'instruments' are mentioned, but the reader are not introduced til the instruments yet. Also the same problem line 86-87. 

line 102, how did you assess if patients were a physical and psykological condition to fill in the scales?

the section describing the scales are without any information about why you chose theese scales, the validity, the reliability, the translation procedure etc. What do they measure, how are they build etc. The last sentence (line119) gives no meaning at all.

2.4 Statistical analysis. You do not describe other analyses than the regression. You should describe them all.

3. Results. Participants (line 138-156.) You should make a table for this information. As it is, is too difficult to read.

Line 160-62. What is meant by degree? What does it mean that nurses masters degree is reduced to 11.8%?        

Line 174-79. Why is Cronbach's alpha important - same question with all scales?  

Rest of the result section is not possible for me to evaluate as you have not provided me with sufficient information of the scale - to understand  the results. 

Discussion: In general the discussion is not related to the findings of the study and thus not really setting the results in relation to other studies and theory.

Conclusion. I do not think that this is a conclusion. It is merely a further discussion bringing new knowledge into discussion, which is not appropriate i a conclusion. Among others, explanation of what aspecs of care the scales consists of as far as I understand (line line 392-398.)   

Reviewer 3 Report

Dear Editor

Thank you very much for choosing me as a reviewer. There are some issues to be considered to improve the quality of the paper.

Abstract:

Please check your Keywords with Mesh data base.

Methods:

The sample size is determined based on which statistical formula?

Should the tools be fully explained? How many questions did each tool have? How was the scoring method? Does the tool have a cutting point? What were the minimum and maximum scores?

Should the validity and reliability of the instruments be explained?

Discussion:

Your discussion needs to be more powerful, so don’t worry and make some changes.

Reviewer 4 Report

Dear Authors,

As I understand your article, you aim to show the correlations between the nursing/care environment and patient satisfaction basing on empirical findings on two hospital units (in Portugal). For this purpose, you consider the perspective of the nurses and the patients and apply three validated measuring instruments. For presentation you choose text, tables and a graphical illustration. In the following, please find my comments.

Title: Please adapt the title of the manuscript. It is about patient satisfaction. The term “Outcomes” is too unspecific.

Abstract: Pleasse, revise. The conclusions are actually the results.

Keywords: Please change.

Introduction:

Line 35 to 36: Is this a complete sentence?

Insert a paragraph on the relevance of the topic for the Portuguese context.

Materials and Methods:

Relevant information is missing: research/study context, choice for the respective hospital units, sampling (nurses, patients), …

Line 80 to 82: What is this supposed to mean?

Line 94 to 96: What is this supposed to mean?

Data collection instruments: please, provide more extensive information on the data collection/measure instruments. For that purpose, please shift relevant paragraphs from the “Results” Chapter to the “Materials and Methods” Chapter.

Line 122: redundant (cf. line 70).

Results:

Patients: line 145 to 146: “a mean age of 2.93”??

Nurses: line 158: “randomized sub-sample” – please, explain? (à “Materials and Methods” Chapter)

Sub-chapter 3.3.: please, provide a clear description of the “three dimensions”.  Description for “clinical care” is missing.

Sub-chapter 3.4.: factor 1 and factor 3 both address “quality of care” – please, revise or rather explain.

Figure 1 may be more appropriate for the “Discussion” Chapter and there could set up the framework for argumentation.

General remark: The “Results” Chapter contains results as well as interpretation of results. The latter (e.g. line 202) should be part of the “Discussion” Chapter.

Discussion:

This chapter needs more structure (sub-headings!) and contains much text that – in my opinion – would be more appropriate for the “Introduction” Chapter.

The discussion lacks references to the demographics of the patients. Why? Are they – for instance – not relevant? …

All the best.

Reviewer 5 Report

Nursing Practice Environment and Nurse-Patient Interactions 2 in Outcomes with Patients in a Hospital Context

REVIEW

Thanks for giving me the opportunity to review this manuscript. The topic is interesting and potentially able to contribute to the nursing profession. However, the manuscript is badly written, with problems of scientific writing and methodological presentation. What follows is a series of suggestions to improve the manuscript.

GENERAL

1.  The text needs intense proofreading and a concise and appropriate scientific language.

ABSTRACT

1.  The authors wrote: . The model is statistically significant, with good fit to the data. Please, be more specific, by reporting p value (s) and the fit indices.

2.  The results section is missing. There is no mention of the answers in relation to the aim of the study. What is exactly the relationship that was found?

3.  Organizations involved in health care demonstrate a better response. The rest of the sentence is missing. Compared to what?

4.  A rationale should emerge from the abstract. What is the gap in the literature?

INTRODUCTION

1.  Please modify the first sentence: in Portugal, and in Europe in general..

2.  The gap in the literature is not clear. Please elaborate more on why it is important to conduct such a study, and what is currently not known so far.

3.  The aim should be broadened by reporting more about this relationship. What influences what? What is the outcome?

4.  The names of hospital1_P etc, should be avoided and replaced by the type of units.

5.  What is the conceptual framework or theory on which the authors based their aims and hypotheses?

METHODS

1.  Please better describe the constructs measured by each instrument

2.  In the statistical analysis, the authors claim that they used simple regression models. But I think this is not all. They also wrote about AMOS, so I deduce also SEM or path analyses were also conducted.

3.  Please elaborate more on what the variables of the models are in terms of independent and dependent variables.

4.  Were the variables taken at one time point?

5.  Results are in disorder and not easily understood. The language should be more scientific and rigorous, and less colloquial (e.g., make up).

6.  I expected that Table 1 reported the sociodemographic characteristics of the sample. These variables help in understanding their possible influence on the relationships tested.

7.  I also expected a second table where all the information reported in what are now Table 1, 2, and 3 , should be reported. This is secondary information with respect to the aims of the study.

8.  Please adapt the graphical representation as per international guidelines

9.  Statistical analysis need to be expanded, the use of the SEM described, and the acceptable fit indices. Also how the assumptions were verified.

RESULTS

This section is in disorder, and the language not properly scientific. A more structured and synthetic part is needed. Please also avoid redundant parts. E.g., “The reliability of the PES-NWI scale was evaluated by calculating Cronbach's alpha 174 internal consistency”. Sentences like this are more adapt for the statistical analysis section.

What are exactly the fit indices of the model? Please report them in a more complete and conventional scientific writing (e.g., CIs of RMSEA, chi square etc.)

DISCUSSION

The discussion should begin by repeating the aims of the study, the results and the novelty.

A further evaluation of the Discussion is needed, after the major problems in the presentation of the manuscript has been fixed

I am looking forward to evaluating the next version of this manuscript.

Round 2

Reviewer 4 Report

Dear Authors,

I enjoyed reading the revised version of your manuscript.

All the best!

Author Response

Dear Reviewer

Thank you very much.

Best regards

Reviewer 5 Report

Thanks for addressing my comments.

I have  no other recommendations

Author Response

(The authors gave the same response as above.)
